# Social contact behaviors are associated with infection status for *Trichuris* sp. in wild vervet monkeys (*Chlorocebus pygerythrus*)

**Brandi Wren** [1,2‡]*, **Ian S. Ray** [2,3,4☯], **Melissa Remis** [1‡], **Thomas R. Gillespie** [5‡], **Joseph Camp** [6☯]

1 Department of Anthropology, Purdue University, West Lafayette, Indiana, United States of America,
2 Applied Behavioural Ecology and Ecosystem Research Unit, University of South Africa, Florida, Republic of South Africa, 3 Dr3 Research and Consulting, LLC, Denver, Colorado, United States of America,
4 Department of Research Methods and Statistics, University of Denver, Denver, Colorado, United States of America, 5 Departments of Environmental Sciences and Environmental Health and Program in Population Biology, Ecology, and Evolutionary Biology, Emory University and Rollins School of Public Health, Atlanta, Georgia, United States of America, 6 Department of Comparative Pathobiology, College of Veterinary Medicine, Purdue University, West Lafayette, Indiana, United States of America

☯ These authors contributed equally to this work.
‡ MR and TRG also contributed equally to this work. BW is senior author on this work.
* brandiwren@gmail.com

**Data Availability Statement:** All parasite, behavior, and GPS data files are available from the Open Science Framework database under the title Primate Parasite Project(s) (https://osf.io/m9spk/).

## Abstract

Social grooming in the animal kingdom is common and serves several functions, from removing ectoparasites to maintaining social bonds between conspecifics. We examined whether time spent grooming with others in a highly social mammal species was associated with infection status for gastrointestinal parasites. Of six parasites detected, one (*Trichuris* sp.) was associated with social grooming behaviors, but more specifically with direct physical contact with others. Individuals infected with *Trichuris* sp. spent significantly less time grooming conspecifics than those not infected, and time in direct contact with others was the major predictor of infection status. One model correctly predicted infection status for *Trichuris* sp. with a reliability of 95.17% overall when the variables used were time spent in direct contact and time spent grooming others. This decrease in time spent grooming and interacting with others is likely a sickness behavior displayed by individuals with less energy or motivation for non-essential behaviors. This study emphasizes the possible links between host behavior and parasitic infections and highlights the need for an understanding of a study population's parasitic infections when attempting to interpret animal behavior.

## Introduction

Grooming is widespread in the animal kingdom, from insects [1–4] to rodents [5–8], birds [9–13], and primates [14–17]. Grooming is generally classified into two types: self-grooming, in which individuals groom themselves, and social grooming or allogrooming in which individuals groom others or are groomed by others [18]. Both self-grooming and social grooming have

**Funding:** BW received funding for this study from Wenner-Gren Foundation (grant #7841), L.S.B. Leakey Foundation (grant #203294), Purdue University Dept. of Anthropology, Purdue Research Foundation, and University of South Africa. T.R. Gillespie was funded by Emory University to support laboratory analyses. J. Camp was funded by Purdue University's Dept. of Comparative Pathobiology to support laboratory analyses. The funders had no role in study design, data collection and analysis, decision to publish, or preparation of the manuscript.

**Competing interests:** The authors have declared that no competing interests exist. The private company affiliation does not alter our adherence to PLOS ONE policies on sharing data and materials.

hygienic benefits including removal of parasites and debris [19–21], as well as physiological benefits like releasing endorphins and lowering heart rate [22, 23]. Social grooming specifically also has social, reproductive, and resource acquisition benefits [24–28]. Grooming with others can also be costly, however, because it may lead to transmission of viruses, bacteria, and other pathogens [29–34]. Studies of rodents and primates have even suggested that social grooming may increase likelihood of infection with parasites that are not typically transmitted between hosts, like soil transmitted nematodes for example [35–37].

Grooming and sociality in general have a positive influence on reproductive fitness, in part by increasing overall health. For example, social integration (often measured largely by grooming behaviors) was associated with increased reproductive success in feral mares [38], and female savannah baboons (*Papio cynocephalus*) [39], while allopreening was related to increased reproductive fitness in common guillemots (*Uria aalge*) [40]. Being groomed by others has been shown to reduce heart rate in horses [23], a pigtail macaque (*Macaca* sp.) [41], and rhesus macaques (*Macaca mulatta*) [42]. A study on Barbary macaques (*Macaca sylvanus*) suggested that grooming others can reduce stress, as indicated by assessments of fecal glucocorticoids [43].

Research supports claims that social grooming can help reduce numbers of ectoparasites, often termed the hygiene hypothesis [14]. In one study of Japanese macaques, researchers concluded that the main function of grooming was to eliminate ectoparasites, specifically lice [19]. Another study revealed that wild savannah baboons (*Papio cynocephalus*) that were groomed more frequently had fewer ticks [44]. Further, research suggests that social grooming may reduce mortality risk from the fungus *Metarhizium anisopliae* among termite hosts (*Zootermopsis angusticollis*) [45].

However, this approach does not fully acknowledge the complexity of all host-parasite relationships and the multifaceted relationship between grooming and health. Understanding host-parasite ecology means understanding the complex interplay of a number of factors including distribution of parasites in the environment and likelihood of encountering them, age, sex, physiology, and social behavior [46, 47]. Dunbar [25] concluded that the hygiene hypothesis alone could not account for primate social grooming behaviors because a meta-analysis revealed that amount of time spent grooming with others correlated more with group size than with body size across the Order Primates overall. Dunbar's work suggested that social grooming may serve more of a hygienic function among New World monkeys (among whom grooming time correlated more precisely with body weight than group size) while serving more of a social function among Old World monkeys and apes (among whom grooming time correlated more precisely with group size than body weight). Further, some studies have found that parasite loads do not correlate with grooming behaviors. For example, grooming behaviors in the Seychelles warbler (*Acrocephalus sechellensis*) were not correlated with feather mite load [48]. Observers also noted that chacma baboons (*Papio ursinus*) did not always remove ticks (*Rhipicephalus* sp.) from partners when grooming with them, even when researchers could see from afar that ticks were engorged [49]. It is also worth noting that tick infestations were estimated to cause over half of known infant deaths among that study population.

Some parasites may be more likely to be transmitted when two individuals groom, and it is not uncommon for some types of pathogens, like viruses, to be transmitted through social contact. This is being increasingly acknowledged as societies around the world have dealt with the COVID-19 pandemic, and researchers have noted these connections between social proximity and the spread of infectious disease in both humans and other animal species [50]. Social contact in western lowland gorillas (*Gorilla gorilla gorilla*)–largely observed as grooming–was associated with death from the Ebola-Zaire virus in a study population in Congo [30]. One study on meerkats (*Suricata suricatta*) revealed that those who groomed others more

frequently were more likely to become infected with tuberculosis (*Mycobacterium bovis*) than those that groomed others less frequently [51]. Social grooming in ants (*Lasius* sp.) resulted in transmission of the potentially pathogenic fungus *Metarhizium anisopliae*, however this ultimately aids in developing immunity to the fungus [52]. Examples like this highlight the complexity of the host-parasite relationship and the need for a more nuanced approach to understanding host-parasite dynamics.

The aim of this study was to determine whether grooming behaviors in a highly social mammal species varied with respect to infection status with gastrointestinal parasites. We examined various dimensions of vervet monkey (*Chlorocebus pygerythrus*) grooming behavior, including time spent grooming others, time spent being groomed by others, and time spent in direct contact with others. We tested fecal samples for gastrointestinal parasites, specifically protozoa and helminths. We then statistically analyzed whether individuals who were infected with parasites spent similar relative amounts of time grooming and/or receiving grooming from other individuals. We anticipated that if social grooming or direct social contact facilitates the transmission of any gastrointestinal parasite species in our study population, then those individuals that spend more time grooming with others should be more likely to exhibit infection with some parasites.

## Materials and methods

### Study site and subjects

Data were collected from three social groups of wild vervet monkeys (*Chlorocebus pygerythrus*) at Loskop Dam Nature Reserve (LDNR), South Africa (Fig 1). LDNR is located in the Olifants River Valley within Mpumalanga and Limpopo provinces (25˚25'S, 29˚18'E), and is managed by the Mpumalanga Tourism and Parks Agency (MTPA). The reserve is 225 km$^2$ and surrounds Lake Loskop, a reservoir of 23.5 km$^2$. The reserve encompasses both highveld and bushveld ecological zones, and habitat ranges from open grasslands to dense woodlands [53]. Common woody species throughout the three groups' ranges include a variety of species of *Combretum*, *Acacia*, *Rhus*, *Grewia*, and *Ficus*, as well as *Dichrostachys cinerea*, *Mimusops zeyheri*, and *Olea europa* [54, 55]. Altitude in LDNR ranges from 990–1450 m and the reserve exhibits a highly seasonal climate. Annual rainfall during the study was 914.5 mm, most of which fell between October and January [2009–2010, LDNR, unpublished. *data*]. Average minimum and maximum temperatures during the study period were 13.5˚C and 26.1˚C, respectively [2009–2010, LDNR unpublished. *data*; 2009–2010, Wren unpublished. *data*].

We chose *Chlorocebus pygerythrus* as the study species because individuals exhibit variation in grooming behaviors [56], allowing us to examine differences in the relationship between social behaviors and parasite infection status. Groups of *Ch. pygerythrus* in LDNR–and much of the surrounding region–typically vary in size from 13–25 individuals [54, 55, 57]. Six groups of *Ch. pygerythrus* at LDNR are habituated, and researchers have been conducting studies of these groups semi-regularly for more than a decade [58–61]. We collected data from three of the six habituated groups at LDNR: Blesbok group, Donga group, and Bay group. At the commencement of the study there were 14 individuals in the Blesbok group, 16 in the Donga group, and 17 in the Bay group; the total study population fluctuated due to births, migrations, and deaths, and was 54 at the conclusion of the study. Here we present data on a total of 55 subjects as well as a subset of 38 of those study subjects. Information on group composition for each social group can be found in Wren [55] and Wren et al. [36, 62]. We located groups using known sleeping sites and home ranges. Data were recorded for only the Blesbok group from July–October 2009 because other researchers were studying the Donga and Bay groups during that time. Data were collected from all three social groups for the remainder of the project.

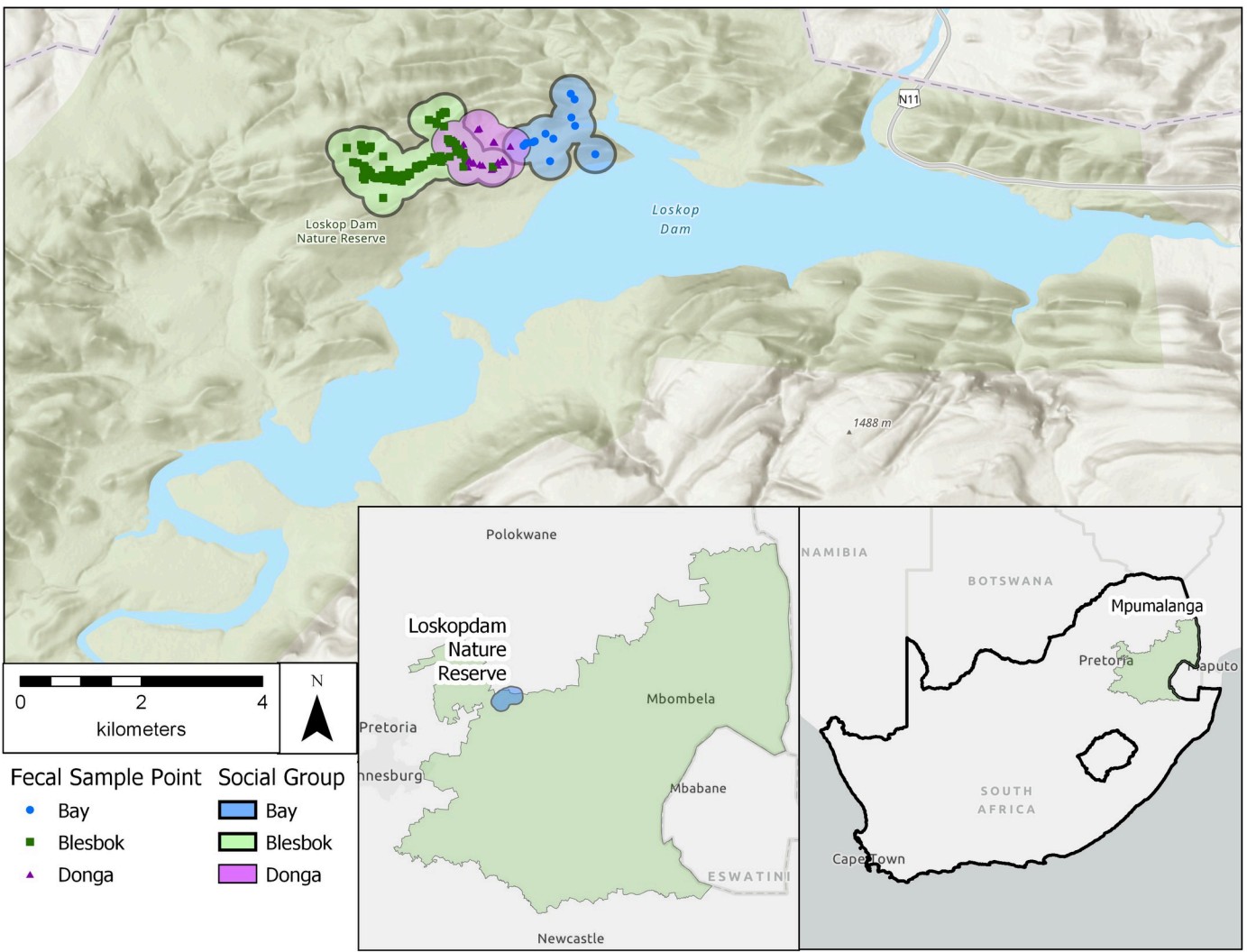

**Fig 1. Approximate home range and fecal sample collection points for wild vervet monkeys at LDNR.**

Individual vervet monkeys were identified using a combination of traits including sex, age class, facial scars, broken limbs or tails, tears and punctures on the ears, physical disabilities, and other traits. When there was any uncertainty over the identity of an individual, those data were excluded from the study. All research assistants had to achieve a 95% interobserver reliability threshold to be allowed to collect data on their own to ensure reliability of the identification of individuals. Further details on identification methods and locating individuals can be found in Wren [55] and Wren et al. [36, 62].

### Behavioral data collection

From July 2009 through July 2010 we followed *Ch. pygerythrus* groups and collected observational data [63]. Although most data were collected between 07:00 h and 16:00 h, observations were conducted between 05:00 h and 19:00 h. We performed 30-min focal follows on each individual. Some follows were terminated early because monkeys were lost from sight, but we kept data from all follows longer than 5 min. We used continuous recording for all behavioral data [63].

We recorded data on the following variables for all bouts of social grooming: start and stop times, whether grooming was given or received, and identity of grooming partner. A grooming bout was considered to be a new bout when any of the following conditions were met: the focal individual stopped grooming or being groomed for 30 seconds or more, the direction of grooming switched (i.e., the individual being groomed began grooming its partner or vice versa), or the individual switched grooming partners. We also recorded data on start and stop times for direct physical contact with another individual and identification of direct social contact partners.

## Fecal sample collection and analysis

We collected 332 fecal samples non-invasively from identified individuals directly following defecation, and samples were immediately preserved in a 10% buffered formalin solution. We recorded data on the following variables for each sample: date, individual, social group, location (GPS coordinates), consistency and color of feces, and whether adult worms were visible in the stool. Fecal samples varied from approximately 3 to 7 g.

We used three methods to detect parasite eggs and cysts in samples in order to reduce the risk of false negatives: fecal flotation, fecal sedimentation, and immunofluorescence microscopy. We isolated helminth eggs and protozoan cysts and oocysts from fecal material using fecal flotation with double centrifugation (at 1800 rpm for 10 min) in $NaNO_3$ solution and fecal sedimentation with dilute soapy water [64]. We also used immunofluorescence microscopy with a Merifluor *Cryptosporidium/Giardia* Direct Immunofluorescent Detection Kit (Meridian Bioscience Inc., Cincinnati, Ohio, USA) to detect *Cryptosporidium* sp. oocysts and *Giardia* sp. cysts [65]. Parasite eggs and cysts were identified by egg or cyst shape, size, color, and contents for flotation and sedimentations, and measurements of eggs and cysts were taken with an ocular micrometer fitted to a compound microscope. For the immunofluorescence microscopy, we scored fecal samples for the presence or absence of *Cryptosporidium* sp. oocysts and *Giardia* sp. cysts.

## Data analysis

Although we collected 511 h of behavioral data and 332 fecal samples from 55 individual vervet monkeys, we analyzed 477.66 h of data and 272 fecal samples from 38 individuals for our final analyses. Some behavioral data were not used in the final analyses because there were no corresponding fecal samples for some study subjects, and vice versa. We present some descriptive statistics for overall group data based on the larger sample.

We calculated measures of parasite infections following Bush et al. [66]. Richness refers to the number of parasite species detected in a host or group. Prevalence refers to the number of hosts infected with a specific parasite species divided by the number of hosts examined.

We did not consider negative samples "sandwiched" between positive samples to be indicative of elimination of an infection nor did we attempt to examine changes in infection status over time. It is widely acknowledged that output of eggs and other stages of gastrointestinal parasites are not always indicative of the intensities of infections [for a discussion see 64]. As such, we did not attempt to examine fecal egg counts or levels of intensity for infections, and we assumed negative samples sandwiched between positive samples were false negatives rather than elimination and reinfection. This follows typical best practices in detecting gastrointestinal parasites in fecal samples which suggests obtaining multiple fecal samples for detection because of false negatives. Currently there are no recorded observations of self-medication in vervet monkeys, so we have no reason to expect self-medication and elimination of infections in our study groups.

We used analysis of variance (ANOVA) to examine whether social groups differed by sampling effort, grooming behaviors, or parasite infections. We planned contrasts to compare each group and each combined grouping of groups to identify statistically significant differences in each observed variable. We further explored significant results from the ANOVA using a logistic regression model from calculated z-scores for each independent variable. We set the presence of *Trichuris* sp. as the binary outcome variable with the following observed variables: time observed, total seconds observed, number of grooming partners, number of grooming partners giving grooming, number of grooming partners receiving grooming, number of total contact partners, time spent giving grooming, time spent receiving grooming, time spent self grooming, time in direct contact, and time spent playing.

We used GNU PSPP 1.2.0 for all statistical tests and QGIS 3.14 for mapping. We set the significance level at *p < 0.05* and considered all tests two-tailed.

### Ethical note

This study was conducted with approvals from LDNR, MTPA, Applied Behavioural Ecology and Ecosystem Research Unit of the University of South Africa, and Purdue University's Animal Care & Use Committee (approval #07–609). We followed all guidelines for the study of nonhuman primates set forth by the International Primatological Society.

## Results

### Descriptive statistics

Mean time observed per individual for the entire study sample, incorporating all three study groups, was 9.3 hours ($n = 55$, minimum = 0.18, maximum = 37.04, SD = 8.84). Age of individuals ranged from 1 to 11 years ($n = 55$, mean = 4.80, SD = 2.67) at the end of the study or last time seen, and 42% of subjects were female (35/55) while 64% were male (35/55).

For parasitological hypothesis testing, we used a subset of the entire study sample that consisted of 38 individuals from across the three social groups. This subset included only individuals for which both behavioral and parasitological data were available and complete. Mean time observed per individual in this subset used for parasitological hypothesis testing was 12.57 hours ($n = 38$, minimum = 1.343, maximum = 37.04, SD = 8.83).

Mean number of fecal samples collected per individual was 7.16 ($n = 38$, minimum = 1, maximum = 25, SD = 8.84). For this subset of 38 individuals for which enough parasitological results were obtained, social groups differed significantly in regard to total time observed ($F_{(2,35)} = 28.242$, $p < 0.001$) and number of fecal samples collected ($F_{(2,35)} = 27.929$, $p < 0.001$). (This was expected, given the data collection schedule reported above.) Age of individuals ranged from 3 to 11 years ($n = 38$, mean = 5.6, SD = 2.29) at the end of the study, and 42% of subjects were female (16/38) while 58% were male (22/38).

### Behavioral results

For the total study sample of 55 individuals, mean proportion of time spent grooming others was 5.0% of total time observed ($n = 55$, mean = 0.05, minimum = 0.0, maximum = 0.29, SD = 0.06). (Table 1) Mean proportion of time spent being groomed for the entire study sample of 55 individuals was 4.0% of total time observed ($n = 55$, mean = 0.04, minimum = 0.0, maximum = 0.01, SD = 0.03). (Table 2) Social groups did not differ in regard to mean time spent grooming others ($F_{(2,52)} = 0.39$, $p = 0.677$) or being groomed by others ($F_{(2,52)} = 0.93$ $p = 0.401$). Time in direct contact, while not grooming, accounted for an average of 20% of the total time observed and did not differ among social groups ($F_{(2,52)} = 0.03$, $p = 0.971$).

**Table 1. Mean percentage of time spent grooming others among vervet monkeys (*Chlorocebus pygerythrus*) at Loskop Dam Nature Reserve, South Africa.**

|  | Infected, Mean Time Observed | Not Infected, Mean Time Observed | t-value | *P*-value |
|---|---|---|---|---|
| *Trichuris* | 4% | 14% | -3.17 | 0.003 |
| Hookworm | 6% | 4% | 0.87 | 0.388 |
| Spirurids | 5% | 6% | -0.90 | 0.375 |
| *Oesophagostomum* sp. | 4% | 9% | -1.74 | 0.090 |
| *Strongyloides* sp. | 3% | 6% | -1.45 | 0.156 |
| *Entamoeba coli* | 3% | 5% | -0.76 | 0.454 |

Degrees of freedom = 36 in all cases.

For the subset of 38 individuals from the study sample used in the parasitological hypothesis testing, mean proportion of time spent being groomed was also 4.0% of total time observed ($n = 38$, mean = 0.04, minimum = 0.0, maximum = 0.01, SD = 0.025). For the subset of 38 individuals, mean proportion of time spent grooming others was 5.1% of total time observed ($n = 38$, mean = 0.05, minimum = 0.0, maximum = 0.29, SD = 0.059). For the subset of 38 individuals, social groups did not differ in regard to mean time spent grooming others ($F_{(2,35)} = 0.172$, $p > 0.05$) or being groomed by others ($F_{(2,35)} = 0.517$, $p > 0.05$). For the subset of 38 individuals, groups also did not differ in mean time spent in direct contact ($F_{(2,35)} = 2.6$, $p > 0.05$).

## Parasitological results

Analyses revealed six types of parasites: *Trichuris* sp. (92% prevalence in the study sample), hookworm (71% prevalence), spirurids (68% prevalence), *Oesophagostomum* sp. (84% prevalence), *Strongyloides* sp. (24% prevalence), and *Entamoeba coli* (92% prevalence) ($n = 38$). Descriptions of these can be found in Wren et al. [35, 53, 60]. We did not detect presence of *Cryptosporidium* sp. or *Giardia* sp. with immunofluorescence microscopy.

Social group differences are presented in Table 3. These differences are likely due to the different sampling efforts for each social group as noted in the methods section. Social groups were significantly different regarding richness of parasite species detected ($F_{(2,35)} = 5.0804$, $p = 0.012$). The Bay group differed significantly from the Donga group ($t(35) = 2.98$, $p = 0.005$) and the Blesbok from the Donga group ($t(35) = 2.58$, $p = 0.014$). The combination of the Bay and Blesbok group differed significantly from the Donga group ($t(35) = 2.05$, $p = 0.048$) while the combination of the Blesbok and Donga group differed significantly from the Bay group ($t(35) = 3.16$, $p = 0.003$).

Social groups differed significantly in the presence of hookworm ($F_{(2,35)} = 1.35$, $p = 0.272$), The Bay group differed from the Blesbok group ($t(35) = 3.45$, $p = 0.001$) and the Donga group

**Table 2. Mean percentage of time spent being groomed among vervet monkeys (*Chlorocebus pygerythrus*) at Loskop Dam Nature Reserve, South Africa.**

|  | Infected, Mean Time Observed | Not Infected, Mean Time Observed | t-value | *P*-value |
|---|---|---|---|---|
| *Trichuris* sp. | 4% | 4% | 0.02 | 0.987 |
| Hookworm | 4% | 4% | 0.80 | 0.430 |
| Spirurids | 4% | 4% | -0.39 | 0.698 |
| *Oesophagostomum* sp. | 4% | 5% | -0.69 | 0.496 |
| *Strongyloides* sp. | 4% | 4% | -0.44 | 0.660 |
| *Entamoeba coli* | 3% | 4% | -0.50 | 0.622 |

Degrees of freedom = 36 in all cases.

**Table 3. Levene's test of homogeneity of variance for behavioural variables in vervet monkeys (*Chlorocebus pygerythrus*) at Loskop Dam Nature Reserve, South Africa.**

| Variable | Levene's Statistic | df1 | df2 | *p* |
|---|---|---|---|---|
| Total Seconds Observed | 6.27 | 2 | 52 | 0.004 |
| Number of Grooming Partners | 5.52 | 2 | 52 | 0.007 |
| Number of Grooming Partners (Giving) | 4.89 | 2 | 52 | 0.011 |
| Number of Grooming Partners (Receiving) | 3.16 | 2 | 52 | 0.051 |
| Number of Total Partners (Contact) | 3.81 | 2 | 52 | 0.029 |
| Time Giving Grooming | 2.68 | 2 | 52 | 0.078 |
| Time Receiving Grooming | 2.90 | 2 | 52 | 0.064 |
| Time Self-Grooming | 8.80 | 2 | 52 | 0.001 |
| Time in Direct Contact | 0.28 | 2 | 52 | 0.756 |
| Time Playing | 4.79 | 2 | 52 | 0.012 |

($t$(35) = 2.44, $p$ = 0.020). The combination of the Bay and Donga groups differed from the Blesbok group ($t$(35) = 2.65, $p$ = 0.012). The combination of the Blesbok and Donga group differed significantly from the Bay group ($t$(35) = 3.34, $p$ = 0.002).

Social groups did not differ significantly in the presence of *Trichuris* sp. ($F_{(2,35)}$ = 1.35, $p$ = 0.272), spirurids ($F_{(2,35)}$ = 2.80, $p$ = 0.075), *Oesophagostomum* sp. ($F_{(2,35)}$ = 0.91, $p$ = 0.412), *Strongyloides* sp. ($F_{(2,35)}$ = 0.17, $p$ = 0.842), or *Entamoeba coli* ($F_{(2,35)}$ = 1.35, $p$ = 0.272).

## Hypothesis testing results

**Social group differences.** One-way analysis of variance (ANOVA) was conducted to examine group differences in observed behaviors. Levene's test of homogeneity of variance revealed only the number of grooming partners, time receiving grooming, and time in direct contact met this assumption (Table 3). However, because ANOVA is robust with respect to violations of homogeneity of variance analyses could still be performed. There were statistically significant differences among groups for: total seconds observed ($F_{(2, 52)}$ = 22.79, $p < 0.001$); number of grooming partners ($F_{(2, 52)}$ = 15.70, $p < 0.001$); number of grooming partners giving ($F_{(2, 52)}$ = 8.11, $p$ = 0.001); number of total partners ($F_{(2, 52)}$ = 19.08, $p < 0.001$); time self-grooming ($F_{(2, 52)}$ = 3.54, $p$ = 0.036) (Table 4).

Planned contrasts revealed specific differences among social groups and combinations of groups (Tables 5 and 6). There were statistically significant differences for all combinations of social groups with respect to total seconds observed. There were statistically significant differences for all combination of social groups with respect to the number of grooming partners except between the combination of the Bay and Blesbok groups compared to the Donga group, ($t$(22.12) = 0.86, $p$ = 0.398). There were statistically significant differences for all combination of social groups with respect to the number of grooming partners giving grooming except between the combination of the Bay and Blesbok groups compared to the Donga group, ($t$(36.79) = 0.63, $p$ = 0.543). There were statistically significant differences for all combination of social groups with respect to the number of grooming partners receiving grooming except between the combination of the Bay and Blesbok groups compared to the Donga group, ($t$(52) = 0.1.03, $p$ = 0.310). There were statistically significant differences for all combination of social groups with respect to the number of total partners except between the combination of the Bay and Blesbok groups compared to the Donga group, ($t$(34.10) = 1.21, $p$ = 0.236). There were no statistically significant differences for any combination of social groups with respect to time giving grooming. There were no statistically significant differences for any combination of social groups with respect to time receiving grooming. There were statistically significant

**Table 4. ANOVA summary for behavioural variables in vervet monkeys (*Chlorocebus pygerythrus*) at Loskop Dam Nature Reserve, South Africa.**

| Variable | SS | df | MS | F | p |
|---|---|---|---|---|---|
| Total Seconds Observed | 25571402466.39 | 2 | 12785701233.20 | 22.79 | <0.001 |
| | 29172783930.95 | 52 | 561015075.60 | | |
| | 54744186397.35 | 54 | | | |
| Number of Grooming Partners | 243.56 | 2 | 121.78 | 15.70 | <0.001 |
| | 403.24 | 52 | 7.75 | | |
| | 646.80 | 54 | | | |
| Number of Grooming Partners (Giving) | 126.13 | 2 | 63.07 | 8.11 | 0.001 |
| | 404.30 | 52 | 7.78 | | |
| | 530.44 | 54 | | | |
| Number of Grooming Partners (Receiving) | 142.37 | 2 | 71.19 | 12.36 | <0.001 |
| | 299.37 | 52 | 5.76 | | |
| | 441.75 | 54 | | | |
| Number of Total Partners (Contact) | 311.25 | 2 | 155.63 | 19.08 | <0.001 |
| | 424.09 | 52 | 8.16 | | |
| | 735.35 | 54 | | | |
| Time Giving Grooming | 0.00 | 2 | 0.00 | 0.39 | 0.677 |
| | 0.20 | 52 | 0.00 | | |
| | 0.20 | 54 | | | |
| Time Receiving Grooming | 0.00 | 2 | 0.00 | 0.93 | 0.401 |
| | 0.04 | 52 | 0.00 | | |
| | 0.04 | 54 | | | |
| Time Self-Grooming | 0.00 | 2 | 0.00 | 3.54 | 0.036 |
| | 0.01 | 52 | 0.00 | | |
| | 0.01 | 54 | | | |
| Time in Direct Contact | 0.01 | 2 | 0.00 | 0.03 | 0.971 |
| | 5.02 | 52 | 0.10 | | |
| | 5.03 | 54 | | | |
| Time Playing | 0.00 | 2 | 0.00 | 1.16 | 0.322 |
| | 0.00 | 52 | 0.00 | | |
| | 0.00 | 54 | | | |

differences for all combinations of social groups with respect to time spent self-grooming except for: the Bay group compared to the Donga group, ($t(32.88) = 0.54$, $p = 0.59$); the combination of the Bay and Blesbok groups compared to the Donga group, ($t(32.62) = -0.79$, $p = 0.435$); the Bay group compared to the combination of the Blesbok and Donga groups, ($t$

**Table 5. Contrast coefficients for ANOVA of behavioural variables in vervet monkeys (*Chlorocebus pygerythrus*) at Loskop Dam Nature Reserve, South Africa.**

| Contrast | Group | | |
|---|---|---|---|
| | 1 (Bay) | 2 (Blesbok) | 3 (Donga) |
| 1 | 1 | 0 | -1 |
| 2 | 1 | -1 | 0 |
| 3 | 0 | 1 | -1 |
| 4 | 1 | 1 | -2 |
| 5 | 1 | -2 | 1 |
| 6 | -2 | 1 | 1 |

**Table 6. ANOVA follow-up results for behavioural variables in vervet monkeys (*Chlorocebus pygerythrus*) at Loskop Dam Nature Reserve, South Africa.**

| Variable | Variance | Contrast | Value of Contrast | *Standard Error* | *t* | *df* | *p*-value |
|---|---|---|---|---|---|---|---|
| Total Seconds Observed | Equal Variance Not Assumed | 1 | -12006.79 | 5271.92 | -2.28 | 28.49 | 0.030 |
| | | 2 | -51396.45 | 8892.15 | -5.78 | 18.99 | <0.001 |
| | | 3 | 39389.66 | 9661.11 | 4.08 | 24.80 | <0.001 |
| | | 4 | 27382.87 | 12774.58 | 2.14 | 36.86 | 0.039 |
| | | 5 | -90786.11 | 17805.10 | -5.10 | 19.12 | <0.001 |
| | | 6 | 63403.24 | 10972.24 | 5.78 | 36.97 | <0.001 |
| Number of Grooming Partners | Equal Variance Not Assumed | 1 | -1.89 | 0.76 | -2.51 | 29.29 | 0.018 |
| | | 2 | -5.17 | 0.96 | -5.41 | 22.10 | <0.001 |
| | | 3 | 3.28 | 1.09 | 3.01 | 30.35 | 0.005 |
| | | 4 | 1.38 | 1.61 | 0.86 | 34.50 | 0.398 |
| | | 5 | -8.45 | 1.90 | -4.43 | 22.12 | <0.001 |
| | | 6 | 7.07 | 1.34 | 5.29 | 48.05 | <0.001 |
| Number of Grooming Partners (Giving) | Equal Variance Not Assumed | 1 | -1.37 | 0.73 | -1.87 | 31.43 | 0.070 |
| | | 2 | -3.72 | 0.99 | -3.76 | 22.31 | 0.001 |
| | | 3 | 2.35 | 1.09 | 2.16 | 28.54 | 0.039 |
| | | 4 | 0.98 | 1.57 | 0.63 | 36.79 | 0.543 |
| | | 5 | -6.07 | 1.95 | -3.12 | 21.37 | 0.005 |
| | | 6 | 5.09 | 1.36 | 3.75 | 46.38 | <0.001 |
| Number of Grooming Partners (Receiving) | Equal Variance Assumed | 1 | -1.26 | 0.78 | 1.62 | 52 | 0.111 |
| | | 2 | -3.92 | 0.80 | 4.9 | 52 | <0.001 |
| | | 3 | 0.66 | 0.80 | 3.32 | 52 | 0.002 |
| | | 4 | 1.40 | 1.36 | 1.03 | 52 | 0.310 |
| | | 5 | -6.58 | 1.40 | 4.70 | 52 | <0.001 |
| | | 6 | 5.19 | 1.36 | 3.81 | 52 | <0.001 |
| Number of Total Partners (Contact) | Equal Variance Not Assumed | 1 | -1.89 | 0.85 | -2.22 | 32.98 | 0.033 |
| | | 2 | -5.80 | 0.95 | -6.12 | 27.26 | <0.001 |
| | | 3 | 3.91 | 1.06 | 3.70 | 32.50 | 0.001 |
| | | 4 | 2.02 | 1.67 | 1.21 | 34.10 | 0.236 |
| | | 5 | -9.72 | 1.82 | -5.34 | 25.27 | <0.001 |
| | | 6 | 7.70 | 1.46 | 5.27 | 47.60 | <0.001 |
| Time Giving Grooming | Equal Variances Assumed | 1 | 0.01 | 0.02 | 0.62 | 52 | 0.538 |
| | | 2 | 0.02 | 0.02 | 0.85 | 52 | 0.397 |
| | | 3 | -0.01 | 0.02 | 0.25 | 52 | 0.801 |
| | | 4 | 0.01 | 0.04 | 0.21 | 52 | 0.838 |
| | | 5 | 0.02 | 0.04 | 0.63 | 52 | 0.529 |
| | | 6 | -0.03 | 0.04 | 0.86 | 52 | 0.395 |
| Time Receiving Grooming | Equal Variances Assumed | 1 | -0.01 | 0.01 | 1.35 | 52 | 0.182 |
| | | 2 | -0.01 | 0.01 | 0.81 | 52 | 0.419 |
| | | 3 | 0.00 | 0.01 | 0.50 | 52 | 0.620 |
| | | 4 | -0.02 | 0.02 | 1.07 | 52 | 0.291 |
| | | 5 | 0.00 | 0.02 | 0.18 | 52 | 0.857 |
| | | 6 | 0.02 | 0.02 | 1.25 | 52 | 0.216 |

(*Continued*)

**Table 6.** (Continued)

| Variable | Variance | Contrast | Value of Contrast | Standard Error | t | df | p-value |
|---|---|---|---|---|---|---|---|
| Time Self-Grooming | Equal Variances Not Assumed | 1 | 0.00 | 0.01 | 0.54 | 32.88 | 0.590 |
| | | 2 | 0.01 | 0.00 | 2.70 | 19.26 | 0.014 |
| | | 3 | -0.01 | 0.00 | -2.76 | 20.36 | 0.012 |
| | | 4 | -0.01 | 0.01 | -0.79 | 32.62 | 0.435 |
| | | 5 | 0.02 | 0.01 | 3.73 | 38.52 | 0.001 |
| | | 6 | -0.02 | 0.01 | -1.60 | 23.03 | 0.123 |
| Time in Direct Contact | Equal Variances Assumed | 1 | 0.02 | 0.10 | 0.18 | 52 | 0.860 |
| | | 2 | -0.01 | 0.10 | 0.06 | 52 | 0.954 |
| | | 3 | 0.02 | 0.10 | 0.23 | 52 | 0.818 |
| | | 4 | 0.04 | 0.18 | 0.24 | 52 | 0.813 |
| | | 5 | -0.03 | 0.18 | 0.17 | 52 | 0.869 |
| | | 6 | -0.01 | 0.18 | 0.07 | 52 | 0.947 |
| Time Playing | Equal Variances Not Assumed | 1 | 0.00 | 0.00 | 1.35 | 18.22 | 0.192 |
| | | 2 | 0.00 | 0.00 | 0.31 | 27.22 | 0.759 |
| | | 3 | 0.00 | 0.00 | 1.87 | 16.68 | 0.079 |
| | | 4 | 0.00 | 0.00 | 2.07 | 28.23 | 0.048 |
| | | 5 | 0.00 | 0.00 | -0.45 | 33.68 | 0.657 |
| | | 6 | 0.00 | 0.00 | -0.83 | 20.59 | 0.419 |

(23.03) = -1.60, $p$ = 0.123). There were no statistically significant differences for any combination of social groups with respect to the time spent playing except between the combination of the Bay and Blesbok groups compared to the Donga group, ($t$(28.23) = 2.07, $p$ = 0.048).

**Generalized linear model.** All behavioral variables from all observed individuals were converted to z-scores for inclusion in the generation of a multivariate logistic regression equation, with the dependent variable modeled as the presence or absence of *Trichuris* sp. The only individual statistically significant effect was time in direct contact (β = -4.63, $p$ = 0.02; Table 7).

**Table 7. Logistic regression summary table for univariate and equations predicting *Trichuris* infection status in vervet monkeys (*Chlorocebus pygerythrus*) at Loskop Dam Nature Reserve, South Africa.**

| Z-score of Variable | β | SE | Wald | p | Exp(β) |
|---|---|---|---|---|---|
| Multivariate Equation: | | | | | |
| Total Seconds Observed | -2.64 | 1.8 | 2.16 | 1.42 | 0.07 |
| Number of Grooming Partners | -0.21 | 4.69 | 0.00 | 9.65 | 0.81 |
| Number of Grooming Partners Giving | -0.74 | 2.85 | 0.07 | 7.96 | 0.48 |
| Number of Grooming Partners Received | 3.33 | 2.91 | 1.31 | 0.253 | 27.95 |
| Number of Total Partners | 2.58 | 2.66 | 0.94 | 0.332 | 13.14 |
| Time Giving Grooming | -.034 | 2.51 | 0.147 | 0.225 | 0.05 |
| Time Receiving Grooming | 0.14 | 0.77 | 0.03 | 0.858 | 1.15 |
| Time Self Grooming | -0.83 | 0.79 | 1.10 | 0.294 | 0.44 |
| Time in Direct Contact | -4.63 | 1.99 | 5.42 | 0.020 | 0.01 |
| Time Playing | -0.74 | 0.60 | 1.53 | 0.216 | 0.48 |
| Bivariate Equation: | | | | | |
| Number of Total Partners | 0.79 | 0.46 | 2.29 | 0.084 | 2.21 |
| Time in Direct Contact | -2.76 | 0.84 | 10.88 | 0.001 | 0.06 |
| Univariate Equation: | | | | | |
| Time in Direct Contact | -4.63 | 1.99 | 5.42 | 0.020 | 0.01 |

**Table 8. Logistic regression fit statistics for univariate and multivariate equations predicting *Trichuris* infection status in vervet monkeys (*Chlorocebus pygerythrus*) at Loskop Dam Nature Reserve, South Africa.**

| Statistic | Equation | Value |
|---|---|---|
| -2 Log Likelihood | Univariate | 37.68 |
| | Bivariate | 34.11 |
| | Multivariate | 24.22 |
| Cox and Snell pseudo-$R^2$ | Univariate | 0.37 |
| | Bivariate | 0.43 |
| | Multivariate | 0.55 |
| Nagelkerke pseudo-$R^2$ | Univariate | 0.50 |
| | Bivariate | 0.57 |
| | Multivariate | 0.73 |

The model shows excellent fit based on the -2 Log likelihood (24.22) and Nagelkerke pseudo-$R^2$ (0.73) (Table 8). This equation was able to correctly predict 100% of observed absence of *Trichuris* sp. and 94.44% of observed presence of *Trichuris* sp., for a total of 95.17% of cases correctly predicted (Table 9).

A bivariate logistic regression was conducted using only the z-score of time in direct contact and the total number of grooming partners to predict the presence of *Trichuris* sp. This regression indicated that time in direct contact with others had a statistically significant negative effect on infection status ($\beta$ = -2.76, *p* = 0.001). Total number of grooming partners had a non-statistically significant positive effect on infection status ($\beta$ = 0.79, *p* = 0.084). The model fit for this equation is worse than for the multivariate equation, based on a higher -2 Log likelihood (34.11) and lower Nagelkerke pseudo-$R^2$ (0.73) (Table 8). This equation was able to correctly predict 60% of observed absence of *Trichuris* sp. and 94.44% of observed presence of *Trichuris* sp., for a total of 90.24% of cases correctly predicted (Table 9).

A univariate logistic regression was conducted using only the z-score of time in direct contact and the presence of *Trichuris* sp. This regression indicated that time in direct contact with others had a statistically significant effect on infection status ($\beta$ = -2.73, *p* = 0.001). The model fit for this equation is worse than for the multivariate equation, based on a higher -2 Log likelihood (37.68) and lower Nagelkerke pseudo-$R^2$ (0.50) (Table 8). This equation was able to correctly predict 60% of observed absence of *Trichuris* sp. and 86.11% of observed presence of *Trichuris* sp., for a total of 82.93% of cases correctly predicted (Table 9).

**Table 9. Logistic regression classification by univariate and multivariate equations for predicting *Trichuris* infection status in vervet monkeys (*Chlorocebus pygerythrus*) at Loskop Dam Nature Reserve, South Africa.**

| | | Predicted Present | Predicted Absent | Percent Correctly Predicted |
|---|---|---|---|---|
| Multivariate | Observed Present | 5 | 0 | 100 |
| | Observed Absent | 2 | 34 | 94.44 |
| | Total | 7 | 34 | 95.12 |
| Bivariate | Observed Present | 3 | 2 | 60 |
| | Observed Absent | 2 | 34 | 94.44 |
| | Total | 5 | 36 | 90.24 |
| Univariate | Observed Present | 3 | 2 | 60.00 |
| | Observed Absent | 5 | 31 | 86.11 |
| | Total | 8 | 33 | 82.93 |

## Discussion

Vervet monkeys at LDNR that are infected with *Trichuris* sp. tend to spend significantly less time grooming conspecifics as well as less time in direct contact with others when compared to those that are not infected. Infected individuals at LDNR spent an average of 4% of their observed time grooming others, while those not infected with this parasite spent an average of 14% of their observed time grooming others. However, no differences existed in time spent being groomed by others. Overall, for the entire study sample (*n* = 55), study subjects spent about 20% of their time in direct contact with another individual. The subset used for parasito-logical analysis spent 6.5% (*n* = 38) of their time in direct contact with another individual. This large difference is primarily influenced by the inclusion of infants and mothers with infants in the entire sample of *n* = 55, but only mothers in the subset with parasitological results of *n* = 38. These mother-infant dyads remain in almost constant contact for the first weeks of a monkey's life and this inflates the overall mean for the group. Because there were not enough fecal samples from these infants, their behavioral data was not included in hypothesis testing. One model that we built was able to correctly predict infection status for *Trichuris* sp. with a reliability of 95.17% overall, but the major factor for prediction was time spent in direct contact.

Although we anticipated results that support the hypothesis that social grooming facilitates transmission of some gastrointestinal parasites, these did not. One possible explanation is that individuals that are infected with *Trichuris* sp. experience degraded health and/or less motiva-tion to groom others and interact with others. Red colobus monkeys (*Procolobus rufomitratus*) in Uganda that were infected with *Trichuris* sp. decreased their time spent performing a num-ber of behaviors, including grooming others [67]. Those same individuals spent more time resting as well as ingesting plant species and/or parts that suggest self-medicative behavior. Whipworm is known to cause anemia, chronic dysentery, rectal prolapse, and poor growth in humans with symptomatic infections [68], so less energy, motivation, or interest for behaviors like social grooming should not be surprising in other species.

Another possible explanation is that *Trichuris* sp. more directly alters host behavior in ver-vet monkeys. Gastrointestinal parasites are known to alter host behavior in some host-parasite relationships, an idea referred to as the manipulation hypothesis [69–72]. For example, *Toxo-plasma gondii* causes intermediate rodent hosts to be more attracted to the scent of felid preda-tors, which are also the definitive host for the parasite [73]. *Dicrocoelium dendriticum* causes infected ants to wait on the tips of blades of grass where they can be ingested by sheep, the par-asite's definitive host. Because manipulation of host behavior usually serves to facilitate trans-mission of the parasite from an intermediate host to a definitive host, and vervet monkeys do not serve as intermediate hosts for *Trichuris* sp., the manipulation hypothesis does not ade-quately explain the results of this study.

Other studies have found multiple morphotypes of *Trichuris* sp. in nonhuman primate hosts in captivity in Nigeria [74, 75], suggesting that potentially multiple species of *Trichuris* sp. may infect nonhuman primates. The major implication of this has been seen as relevant for public health because it may mean that the species of *Trichuris* sp. that infect humans and non-human primates are not the same, suggesting that transmission of *Trichuris* sp. between humans and other primates is not as severe a public health concern as previously considered. However, it could also have implications for how primate hosts respond to or become infected with *Trichuris* sp.

Hart [76] noted that ill or infected animals display altered behavior, and argued that these sickness behaviors can be adaptive. One study of chimpanzees (*Pan troglodytes schweinfurthii*) revealed that infected individuals exhibit altered behavior, most fittingly described as lethargy

[77]. Behavioral changes due to parasitic infections in fish have been observed and range from mating behaviors to foraging efficiency (reviewed in Barber et al. [78]). The Ghai et al. [67] study that revealed that *Trichuris* sp. was associated with a reduction in grooming and mating and also found that individuals infected with this parasite took longer to switch behaviors than those individuals that were not infected. These studies increasingly suggest that host-parasite dynamics have far-reaching consequences for animal behavior.

This study suggests that the gastrointestinal parasite *Trichuris* sp. is associated with behavioral differences, specifically decreased time spent grooming others and time spent in direct contact with others, in vervet monkey hosts. These behavioral differences are extreme enough to influence group means when assessing behavior. Further, if an individual is less likely to groom or interact with conspecifics, then they may also experience lower social status and thus lower reproductive fitness. These results highlight the need for parasitological analyses for a complete and nuanced understanding of animal behavior.

## Acknowledgments

We would like to acknowledge the contributions of our field research assistants Dr. Shawn Hurst, Andrew Kalmbach, Katie Dean, Claire Hammy, Liz Sperling, Lisa Breland, Michele Mignini, and Ruby Malzoni. We would like to thank the staff at LDNR, including Jannie Coetzee, as well as the UNISA staff, including Dr. Leslie Brown, Ms. Tersia De Beer, and Dr. Alan Barrett. We would also like to thank Elizabeth Canfield, Naomi Hauser, and Michele Moses for their assistance in the fecal analysis at Emory University.

## Author Contributions

**Conceptualization:** Brandi Wren, Melissa Remis, Thomas R. Gillespie.

**Data curation:** Brandi Wren, Ian S. Ray.

**Formal analysis:** Brandi Wren, Ian S. Ray.

**Funding acquisition:** Brandi Wren, Melissa Remis, Thomas R. Gillespie.

**Investigation:** Brandi Wren, Ian S. Ray, Thomas R. Gillespie, Joseph Camp.

**Methodology:** Brandi Wren, Melissa Remis, Thomas R. Gillespie, Joseph Camp.

**Project administration:** Brandi Wren, Melissa Remis, Thomas R. Gillespie, Joseph Camp.

**Resources:** Brandi Wren, Melissa Remis, Thomas R. Gillespie, Joseph Camp.

**Software:** Brandi Wren, Ian S. Ray.

**Supervision:** Melissa Remis, Thomas R. Gillespie, Joseph Camp.

**Validation:** Ian S. Ray.

**Visualization:** Ian S. Ray.

**Writing – original draft:** Brandi Wren.

**Writing – review & editing:** Brandi Wren, Ian S. Ray, Melissa Remis, Thomas R. Gillespie, Joseph Camp.

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
