## [Decision Letter · Decision Letter 0]

8 Dec 2020

PONE-D-20-30455

Social contact and grooming predicts infection status with whipworm (*Trichuris* sp.) in wild vervet monkeys (*Chlorocebus pygerythrus*)

PLOS ONE

Dear Dr. Wren,

Thank you for submitting your manuscript to PLOS ONE. After careful consideration, we feel that it has merit but does not fully meet PLOS ONE’s publication criteria as it currently stands. Therefore, we invite you to submit a revised version of the manuscript that addresses the points raised during the review process.

The Ms presents an interesting study which is certainly timely. However, The paper lacks precision and clarity in several aspects. Mostly, the methods section must be rewritten specifying how individuals were identified and their behaviour registered, as well as how the dynamics of infection was evaluated. There is a potential confusion about what is considered causes and consequences of behaviour which should be adressed and explained clearly in the results and discussion. The introduction should be revised. Please detail how you adressed the referee's comments in the rebuttal letter.

We look forward to receiving your revised manuscript.

Kind regards,

Nicolas Chaline

Academic Editor

PLOS ONE

Journal Requirements:

We note that one or more of the authors are employed by a commercial company:Dr3 Research and Consulting, LLC.

2.1. Please provide an amended Funding Statement declaring this commercial affiliation, as well as a statement regarding the Role of Funders in your study. If the funding organization did not play a role in the study design, data collection and analysis, decision to publish, or preparation of the manuscript and only provided financial support in the form of authors' salaries and/or research materials, please review your statements relating to the author contributions, and ensure you have specifically and accurately indicated the role(s) that these authors had in your study. You can update author roles in the Author Contributions section of the online submission form.

2.2. Please also provide an updated Competing Interests Statement declaring this commercial affiliation along with any other relevant declarations relating to employment, consultancy, patents, products in development, or marketed products, etc.  

3. We noted in your submission details that a portion of your manuscript may have been presented or published elsewhere.

"The parasite data used in this study were used in two other published studies, listed below. In those studies, these parasite data were examined in relation to other variables not addressed in this current manuscript, specifically parasite infection patterns as they relate to number of grooming partners, social group membership, age, sex, precipitation, and temperature.  

Wren BT, Remis MJ, Camp JW, Gillespie TR. Number of Grooming Partners Is Associated with Hookworm Infection in Wild Vervet Monkeys (Chlorocebus pygerythrus). Folia Primatol. 2016;87:168-179.

Wren BT, Gillespie TR, Camp JW, Remis MJ. Helminths of Vervet Monkeys, Chlorocebus aethiops, from Loskop Dam Nature Reserve, South Africa. Comp Parasitol. 2015;82:101-108."

Please clarify whether this publication was peer-reviewed and formally published. If this work was previously peer-reviewed and published, in the cover letter please provide the reason that this work does not constitute dual publication and should be included in the current manuscript.

5. We note that Figure 1 in your submission contain map images which may be copyrighted. All PLOS content is published under the Creative Commons Attribution License (CC BY 4.0), which means that the manuscript, images, and Supporting Information files will be freely available online, and any third party is permitted to access, download, copy, distribute, and use these materials in any way, even commercially, with proper attribution. For these reasons, we cannot publish previously copyrighted maps or satellite images created using proprietary data, such as Google software (Google Maps, Street View, and Earth). For more information, see our copyright guidelines: http://journals.plos.org/plosone/s/licenses-and-copyright.

5.1.    You may seek permission from the original copyright holder of Figure 1 to publish the content specifically under the CC BY 4.0 license. 

5.2.    If you are unable to obtain permission from the original copyright holder to publish these figures under the CC BY 4.0 license or if the copyright holder’s requirements are incompatible with the CC BY 4.0 license, please either i) remove the figure or ii) supply a replacement figure that complies with the CC BY 4.0 license. Please check copyright information on all replacement figures and update the figure caption with source information. If applicable, please specify in the figure caption text when a figure is similar but not identical to the original image and is therefore for illustrative purposes only.

6. Please include a caption for figure 1.

Reviewers' comments:

Reviewer's Responses to Questions

**Comments to the Author**

1. Is the manuscript technically sound, and do the data support the conclusions?

Reviewer #1: No

2. Has the statistical analysis been performed appropriately and rigorously? 

Reviewer #1: No

3. Have the authors made all data underlying the findings in their manuscript fully available?

Reviewer #1: Yes

4. Is the manuscript presented in an intelligible fashion and written in standard English?

Reviewer #1: Yes

5. Review Comments to the Author

Reviewer #1: In this manuscript, the authors examine correlations between infection status of gastrointestinal parasites and allogrooming given and received. They document differences in infection status with one of these parasites (whipworm) with time spent grooming others, which they suggest could be a consequence of sickness behavior.

This paper is both timely and interesting, and the authors have a large dataset with which to work. However, I had difficulty following their analyses. Some seemed unnecessary, whereas other (necessary) analyses appeared missing.

One major question/ comment that I had was of timing. What was the timing of infection relative to the behavioral data? Did they limit the behavioral data that they used to a certain window of time (e.g., 5 days? 7 days?) before or after feces were collected, so that they could be sure to relate infection status to specific behaviors? Did infection status of individuals change throughout the study, so that they moved from "infected" to "uninfected" categories? It also seems as though they needed to include "individual ID" as a random effect, as bouts of grooming from specific individuals were not independent data points, because individuals could (presumably) change from infected to uninfected status, and because they had substantial variation in the amount of time spent observing specific individuals.

Also, what about co-infections and burden? This might be getting too complicated for a single manuscript, but (if so) the authors should at least acknowledge how these might affect their results.

Major comments:

Introduction:

Very thorough introduction, but overly long and should be shortened to minimize tangential information. Some paragraphs were redundant and could be eliminated/ combined. For example, L127-132 discusses transmission of parasites via grooming that are not typically transmitted by host-to-host transmission. These references could simply be embedded in the last line of the first paragraph (L66-68) which makes a similar point (and which is tangential to this paper, anyway).

Aims and hypothesis, L 141-151. The stated Aim is to "determine whether grooming behaviors.... varied with respect to infection status," testing whether "individuals who were infected.... spent similar amounts of time grooming and/or receiving grooming from other individuals." From these statements, it sounds like the authors are interested in how infection affects grooming behavior (e.g., grooming behavior as the response, infection as predictor). I was therefore confused when their analyses focused on infection as the response and grooming behaviors as the predictors (L236-241). They do state in the final sentence of the Introduction that they predict that individuals that spend more time grooming others should be more likely to be infected (infection as response). However, their conclusions (e.g., that sickness behaviors reduces the amount of allogrooming given among infected individuals; L429-431) are arguing for a "amount of grooming given" as a response, not a predictor of infection.

At any rate, the authors should make clear the direction of the effect that they are testing. If they are testing allogrooming as cause AND effect of infection, then they should clearly explain this in their analytical methods and results and run the analyses accordingly.

Methods and Results.

Study site, L161-169. Data on vegetation and climate seem unnecessary to this paper.

Subjects, L170-185. I am assuming that the researchers were able to identify individuals (which would be key to linking feces to individual behavior), but they do not state this explicitly. Can you provide information on how individuals were recognized/ identified?

Data analysis and results. The authors often provide a total sample size vs. sample size actually used/ analysed (e.g., L254-270). I found this presentation confusing/ unnecessary. Can the authors simply present the results from the subset of data that they used? They might include one sentence explaining the total sample size and why they focused on a subset (e.g., because they had both behavioral and fecal data; L225-226), but not repeatedly throughout.

L236-241: What happened to the other infections --- why do you immediately focus on whipworm? I think that some critical preliminary analyses are missing here (e.g., where you show that there is no correlation between these other parasites and behavior). This was stated in the abstract (44-46) but nowhere else that I could find. And--- as above--- why use infection as the response, if you are interested in how infection affects time spent grooming?

L301- 372; Tables 3-6. A large amount of text is dedicated to differences among groups. This seems tangential and unnecessary to the paper. All of this information could be summarized in a sentence or two and given in the Supplementary materials. Differences among parasites vs. behavior are much more important (essential) to this paper, but are completely absent.

L374. Again, why go straight to whipworm? What happened to the other parasites? And why look at whipworm only as response and not predictor?

Table 7. "Bivariate equation," and L392-401. I am not sure why the authors included this model. "Time in direct contact" was significant in the full model, so that makes sense in the reduced model, but "Number of Total Partners" was not --- so why include it in a bivariate model? The only other term from the full model that looks like it should be in the bivariate model is "time giving grooming," but it is not included here. Please explain choice of terms.

6. PLOS authors have the option to publish the peer review history of their article (what does this mean?). If published, this will include your full peer review and any attached files.

Reviewer #1: No

---

## [Author Response · Author response to Decision Letter 0]

5 Feb 2021

Replies to Editor Comments:

 1. “However, The paper lacks precision and clarity in several aspects. Mostly, the methods section must be rewritten specifying how individuals were identified and their behaviour registered, as well as how the dynamics of infection was evaluated.” 

 a. Where specific cases have been clearly noted by the editor and reviewer, we have attempted to be more precise and clear. 

 b. Although citations detailing the study methods more closely were previously provided, additional information about how individuals were identified has been added to the last paragraph of the study site and subjects section. (lines 186-193)

 c. We are unclear as to what more detail would be desired as far as how behaviour was “registered,” as full detail on the behavioural variables that were studied is included. If the editor could note specifically what detail they would like included, we would be happy to include it. 

 d. Infection dynamics were not the focus of this study, but rather the behavioural correlates of infection were of interest. We are unclear as to what is meant by the phrase “how the dynamics of infection was evaluated.” All methods used for parasitological analyses are fully described in the methods section, with further citations provided for more detail.

 2. “There is a potential confusion about what is considered causes and consequences of behaviour which should be adressed and explained clearly in the results and discussion.”

 a. To clarify, because there does seem to be confusion, this paper examines the differences in behaviours of vervet monkeys infected with whipworm or not infected with whipworm. The analyses here highlight a statistically significant association between infection status and social contact behaviours, and we then suggest an explanation for this association. 

 3. “The introduction should be revised.” 

 a. The only specific comments about the introduction were that it needs to be shorter, so we have edited it down to a shorter length. 

Replies to General Comments:

 a. We have renamed our image file to follow PLOS ONE conventions and uploaded the new file to the Editorial Manager. We have reviewed the PLOS ONE style requirements again and double checked the formatting of the manuscript throughout. 

 2. Thank you for stating the following in the Competing Interests section: "The authors have declared that no competing interests exist." We note that one or more of the authors are employed by a commercial company:Dr3 Research and Consulting, LLC.

 a. The private company did not provide any support or contributions to this study. The data analysis was conducted outside of company time and using personal resources. However, as that author is a 49% owner of the company conducting research and statistical analyses for clients, it would be disingenuous to not list this affiliation.

 3. Please provide an amended Funding Statement declaring this commercial affiliation, as well as a statement regarding the Role of Funders in your study. If the funding organization did not play a role in the study design, data collection and analysis, decision to publish, or preparation of the manuscript and only provided financial support in the form of authors' salaries and/or research materials, please review your statements relating to the author contributions, and ensure you have specifically and accurately indicated the role(s) that these authors had in your study. You can update author roles in the Author Contributions section of the online submission form. . . If your commercial affiliation did play a role in your study, please state and explain this role within your updated Funding Statement.

 a. The private company, Dr3 Research and Consulting, LLC, did not provide any support or contributions to this study. The data analysis was conducted outside of company time and using personal resources.

 4. Please also provide an updated Competing Interests Statement declaring this commercial affiliation along with any other relevant declarations relating to employment, consultancy, patents, products in development, or marketed products, etc. Within your Competing Interests Statement, please confirm that this commercial affiliation does not alter your adherence to all PLOS ONE policies on sharing data and materials by including the following statement: "This does not alter our adherence to PLOS ONE policies on sharing data and materials.” (as detailed online in our guide for authors http://journals.plos.org/plosone/s/competing-interests) . If this adherence statement is not accurate and there are restrictions on sharing of data and/or materials, please state these. Please note that we cannot proceed with consideration of your article until this information has been declared.

 i. We have included updated statements above in the cover letter.

 5. We noted in your submission details that a portion of your manuscript may have been presented or published elsewhere. Please clarify whether this publication was peer-reviewed and formally published. If this work was previously peer-reviewed and published, in the cover letter please provide the reason that this work does not constitute dual publication and should be included in the current manuscript. 

 a. This paper is based on part of a larger project on social behaviour and primate-parasite ecology. Other portions of this larger study have been published in two peer reviewed scholarly journals and in one edited volume, listed below. We have addressed this in the cover letter. 

 i. Peer reviewed publications:

 1. Wren BT, TR Gillespie, JW Camp, and MJ Remis. 2015. Helminths of Vervet Monkeys, Chlorocebus aethiops, from Loskop Dam Nature Reserve, South Africa. Comparative Parasitology 82(1):101-108.

 2. Wren BT, MJ Remis, JW Camp, and TR Gillespie. 2016. Number of Grooming Partners is Associated with Hookworm Infection in Wild Vervet Monkeys (Chlorocebus aethiops). Folia Primatologica 87(3):168-179. doi:10.1159/000448709

 ii. Edited volume publication:

 1. Wren, BT. 2019. Biological Complexity in Primate Sociality and Health. In TR Turner, C Schmitt, & J Danzy Cramer (Eds.), Savanna Monkeys: The Genus Chlorocebus. (pp. 133-140). Cambridge: Cambridge University Press. doi:10.1017/9781139019941.011

 6. Please amend your list of authors on the manuscript to ensure that each author is linked to an affiliation. Authors’ affiliations should reflect the institution where the work was done (if authors moved subsequently, you can also list the new affiliation stating “current affiliation:….” as necessary).

 a. We request the editor provide specifics on what needs to be updated here. All authors are listed with their correct affiliations.

 7. We note that Figure 1 in your submission contain map images which may be copyrighted. All PLOS content is published under the Creative Commons Attribution License (CC BY 4.0), which means that the manuscript, images, and Supporting Information files will be freely available online, and any third party is permitted to access, download, copy, distribute, and use these materials in any way, even commercially, with proper attribution. For these reasons, we cannot publish previously copyrighted maps or satellite images created using proprietary data, such as Google software (Google Maps, Street View, and Earth).

 a. This map was created specifically for this manuscript using QGIS open-source software. This software is governed by creative commons license.

 8. You may seek permission from the original copyright holder of Figure 1 to publish the content specifically under the CC BY 4.0 license. 

 a. The copyright holder is Ian S. Ray, one of the co-authors of this publication who created this map specifically for this work. He grants permission for its use. 

 9. If you are unable to obtain permission from the original copyright holder to publish these figures under the CC BY 4.0 license or if the copyright holder’s requirements are incompatible with the CC BY 4.0 license, please either i) remove the figure or ii) supply a replacement figure that complies with the CC BY 4.0 license. Please check copyright information on all replacement figures and update the figure caption with source information. If applicable, please specify in the figure caption text when a figure is similar but not identical to the original image and is therefore for illustrative purposes only.

 a. The data underlying QGIS comes from these sources that you recommend we use. The software used to create the figure is governed by creative commons license. The figure is not published in any other source. One of the co-authors of the paper is the copyright holder for the image. We are unclear on what the problem with the figure is. 

 10. Please include a caption for figure 1.

 a. We have included a caption for Fig. 1, as highlighted on lines 167-168. (Fig 1. Approximate home range and fecal sample collection points for wild vervet monkeys at LDNR.)

Replies to Reviewer Comments:

 1. One major question/ comment that I had was of timing. What was the timing of infection relative to the behavioral data? Did they limit the behavioral data that they used to a certain window of time (e.g., 5 days? 7 days?) before or after feces were collected, so that they could be sure to relate infection status to specific behaviors?

 a. We did not limit the behavioural data to a specific window of time for a number of reasons. First, we chose to look at means of time spent grooming or in social contact as well as overall infection status. We realize this is a different approach from the Ghai et al. 2015 paper which looked at only data from individuals whose infection status changed throughout the study; our study examined behavioural correlates of infection status, rather than changes in behavior due to infection. Our study, although similarly a study on host-parasite behavioural interactions, did not intend to replicate the results of Ghai et al. In fact, by using a different approach but still obtaining similar results, we believe this further highlights the significance of these two studies together. Essentially, our study corroborates the findings of the Ghai et al. study but does not, and never did, attempt to replicate it. As such, we justify our approach by the following:

 i. We did not choose to restrict behavioural data analysis with reference to positive parasite fecal samples due to the likelihood of false-negative test results. Although there is often mention of individual hosts’ infection status changing throughout a study, there is little evidence that the infection was eliminated rather than simply not detected or displayed in one specific fecal sample. In fact, Ghai et al. (2015) stated, “while positive samples conclusively indicate whipworm infection, negative samples may indicate uninfected individuals or infected individuals that are not actively shedding eggs” (page 2). The only cases in which we have evidence to suggest that infections were eliminated are in studies of self-medication, and there are no recorded or anecdotal observations of vervet monkeys engaging in self-medication. (In fact, in preliminary analyses of the data from this study, we have found no evidence of self-medication in our study population.) To reliably show elimination of infection, an individual would need to show repeated negative fecal samples in the study results. Our study subjects showed positive and negative samples interspersed, suggesting that even if they were eliminating the infection it was only briefly because they were re-acquiring the infection in a matter of days. Further, there is a large body of literature acknowledging that shedding of eggs is highly variable and that negative fecal samples do not always mean there is no infection. (And in fact, when these negative fecal samples are bracketed by positive fecal samples, it suggests these are false negatives rather than eliminated infections.) Thus, overall we chose to work with behavioural data collected from the entirety of the year because it provided a larger sample size to work with while still accurately reflecting infection status of study subjects. Overall, best practices in detecting infections with gastrointestinal parasites suggest multiple fecal samples because of false negatives. One discussion on this can be found in:

 1. Knopp S, Mgeni AF, Khamis IS, Steinmann P, Stothard JR, Rollinson D, Marti H, Utzinger J. Diagnosis of soil-transmitted helminths in the era of preventive chemotherapy: effect of multiple stool sampling and use of different diagnostic techniques. PLoS Negl Trop Dis. 2008 Nov 4;2(11):e331.

 2. Did infection status of individuals change throughout the study, so that they moved from "infected" to "uninfected" categories?

 a. Our study examined behavioural correlates of infection status, rather than changes in behavior due to infection. Due to the complexities of collecting data on wild individuals, we could not ensure that the absence of infection in one fecal sample was representative of the individual's microbiome. Because of the nature of studies using noninvasive fecal samples, we cannot be sure that individuals moved from infected to uninfected status at a given point unless no more positive samples were detected after that time. Please see our discussion about false negatives above. 

 3. It also seems as though they needed to include "individual ID" as a random effect, as bouts of grooming from specific individuals were not independent data points, because individuals could (presumably) change from infected to uninfected status, and because they had substantial variation in the amount of time spent observing specific individuals.

 a. We are unclear as to what the reviewer means when they say that the bouts of grooming are not independent data points because individuals could change from infected to uninfected. If the reviewer could clarify what they mean by this, we would be happy to address it further. Currently as it reads, it is stating that changing infection status is what determines independence of data points. Changing of infection status is problematic for the reasons stated previously regarding false negatives.

 b. Individual identity was used to match behavior and fecal samples, but individual variation in time spent grooming was not modeled. Ultimately, we did not create a multilevel structural equation model incorporating behaviours nested within individuals, nested within groups. Instead, we compared the proportion of time spent completing different activities for the group of infected v. the group of uninfected individuals. We did not model possible infection change over time as previously stated.

 c. Ultimately, some of these comments make the assumption that we are modeling from a population, rather than a sample. (Difference between in vitro analyses and in vivo analyses)

 4. Also, what about co-infections and burden? This might be getting too complicated for a single manuscript, but (if so) the authors should at least acknowledge how these might affect their results.

 a. An examination of co-infections was not the focus of this manuscript and we believe that topic would be beyond the scope of this paper. We were not attempting to examine parasite burden or intensity of infection so that was not discussed.

 b. We have added a discussion of parasite burden/fecal egg output, timing of infections, elimination of infections, and more clarification to the statistical methods/hypotheses where possible to clarify these issues. 

 5. Very thorough introduction, but overly long and should be shortened to minimize tangential information. Some paragraphs were redundant and could be eliminated/ combined. For example, L127-132 discusses transmission of parasites via grooming that are not typically transmitted by host-to-host transmission. These references could simply be embedded in the last line of the first paragraph (L66-68) which makes a similar point (and which is tangential to this paper, anyway). 

 a. We have shortened the introduction and moved the citations referenced in this comment accordingly. 

 6. Aims and hypothesis, L 141-151. The stated Aim is to "determine whether grooming behaviors.... varied with respect to infection status," testing whether "individuals who were infected.... spent similar amounts of time grooming and/or receiving grooming from other individuals." From these statements, it sounds like the authors are interested in how infection affects grooming behavior (e.g., grooming behavior as the response, infection as predictor). I was therefore confused when their analyses focused on infection as the response and grooming behaviors as the predictors (L236-241). They do state in the final sentence of the Introduction that they predict that individuals that spend more time grooming others should be more likely to be infected (infection as response). However, their conclusions (e.g., that sickness behaviors reduces the amount of allogrooming given among infected individuals; L429-431) are arguing for a "amount of grooming given" as a response, not a predictor of infection.

 a. This study is ultimately focused on the interplay between infection status and behaviour. We acknowledge that behaviours influence infections and vice versa, and we discuss this complex interplay in the introduction and literature review. 

 b. This is not the purpose of these analyses. As stated, determining whether grooming behaviors vary with respect to infection status makes no assumption about the direction of effect. For an explanation of how regression can be used as prediction, rather than causation, see: https://statisticalhorizons.com/prediction-vs-causation-in-regression-analysis

 7. At any rate, the authors should make clear the direction of the effect that they are testing. If they are testing allogrooming as cause AND effect of infection, then they should clearly explain this in their analytical methods and results and run the analyses accordingly.

 a. We utilized a framework of group differences (assessed via ANOVA) and correlational relationships between variables. Logistic regression was utilized to determine if we could predict infection status based on observed behaviors, but this was not a causative model. This study is based on correlations, thus the direction of effect is not relevant here and it is not something we tested directly. We do hypothesize on the direction of the relationship as part of our interpretation of the results, however our statistical tests did not examine the direction of this relationship.

 8. Study site, L161-169. Data on vegetation and climate seem unnecessary to this paper.

 a. It is standard to include data and descriptions of the study site for any ecological study, so this comment is very surprising to us. Vegetation is relevant, as vegetation determines what habitat types are available to the monkeys, what foods are available, and ultimately what parasites may be infectious in the environment.

 9. Subjects, L170-185. I am assuming that the researchers were able to identify individuals (which would be key to linking feces to individual behavior), but they do not state this explicitly. Can you provide information on how individuals were recognized/ identified?

 a. Yes, individuals in each study group were identified. A paragraph explaining this has been added to the methods section.

 10. The authors often provide a total sample size vs. sample size actually used/ analysed (e.g., L254-270). I found this presentation confusing/ unnecessary. Can the authors simply present the results from the subset of data that they used? They might include one sentence explaining the total sample size and why they focused on a subset (e.g., because they had both behavioral and fecal data; L225-226), but not repeatedly throughout.

 a. We present descriptive statistics and results using both sets of data to provide the most data available. We have clarified where necessary as to which set of data we are discussing. Presenting data in this manner is typical and we were following precedent from another published study that examined this research topic (Ghai et al., 2015).

 11. L236-241: What happened to the other infections --- why do you immediately focus on whipworm?

 a. Tables 1 & 2 present the results pertaining to the other infections. We focused on whipworm because it was the only gastrointestinal parasite species of those detected that displayed a significant association with grooming and social contact behaviors. 

 12. I think that some critical preliminary analyses are missing here (e.g., where you show that there is no correlation between these other parasites and behavior). This was stated in the abstract (44-46) but nowhere else that I could find.

 a. It is unclear what the reviewer is asking or suggesting here. What is missing? The presentation of no association between other parasites detected and time spent in social grooming behaviors is in Tables 1 & 2 and these analyses are discussed in the associated text. If the reviewer clarifies what they would like added/what is missing, we would be happy to add it.

 13. L301- 372; Tables 3-6. A large amount of text is dedicated to differences among groups. This seems tangential and unnecessary to the paper. All of this information could be summarized in a sentence or two and given in the Supplementary materials. Differences among parasites vs. behavior are much more important (essential) to this paper, but are completely absent.

 a. This information is provided to demonstrate that multiple analyses are not required, as the results indicate there were not statistically significant differences in group behavior. Thus, data could be analyzed in a single omnibus analysis. These analyses are generally considered typical for this analysis. We are unclear what the reviewer is requesting regarding the last sentence in this comment; we presented differences among parasites versus behavior in Tables 1 & 2 and they were not significant.

 14. L374. Again, why go straight to whipworm? What happened to the other parasites? And why look at whipworm only as response and not predictor?

 a. As stated previously, Table 1 presents results of preliminary analyses regarding associations between all parasites detected and time spent on social grooming behaviors. The only parasite species that was found to be significantly associated with the social behavior that was the focus of this study - time spent in social grooming behaviors - was whipworm. The other parasite species had no significant relationships with time spent on social grooming. 

 b. We have addressed the comment re: response versus predictor elsewhere in our response. 

 15. Table 7. "Bivariate equation," and L392-401. I am not sure why the authors included this model. "Time in direct contact" was significant in the full model, so that makes sense in the reduced model, but "Number of Total Partners" was not --- so why include it in a bivariate model? The only other term from the full model that looks like it should be in the bivariate model is "time giving grooming," but it is not included here. Please explain choice of terms.

 a. The fundamental idea we were testing was whether or not grooming was associated with parasite infection. Thus, the number of total grooming partners was considered to be the most representative variable of the diversity of social interactions, whereas time in direct contact was considered the best indicator of the overall time spent in social interaction. The fundamental idea we were testing was whether or not grooming is associated with parasite infection. Because time in Direct Contact includes Time Giving Grooming as a subset of the behaviour, we elected for the broader term for our equation.

Please see cover letter for more clarification.

---

## [Editor Report · Decision Letter 1]

25 Mar 2021

Social contact behaviors are associated with infection status for *Trichuris* sp. in wild vervet monkeys (*Chlorocebus pygerythrus*)

PONE-D-20-30455R1

Dear Dr. Wren,

We’re pleased to inform you that your manuscript has been judged scientifically suitable for publication and will be formally accepted for publication once it meets all outstanding technical requirements.

I am satisfied that you answered the referee's comments adequately and extensively.

Kind regards,

Nicolas Chaline

Academic Editor

PLOS ONE
---

## [Editor Report · Acceptance letter]

29 Mar 2021

PONE-D-20-30455R1 

Social contact behaviors are associated with infection status for *Trichuris* sp. in wild vervet monkeys (*Chlorocebus pygerythrus*) 

Dear Dr. Wren:

I'm pleased to inform you that your manuscript has been deemed suitable for publication in PLOS ONE. Congratulations! Your manuscript is now with our production department. 

Kind regards, 

on behalf of

Professor Nicolas Chaline 

Academic Editor

PLOS ONE